# Coir-Based Growing Media with Municipal Compost and Biochar and Their Impacts on Growth and Some Quality Parameters in Lettuce Seedlings

Tiago Carreira Martins [1], Rui M. A. Machado [1,*], Isabel Alves-Pereira [2], Rui Ferreira [2,*] and Nazim S. Gruda [3]

[1] MED—Mediterranean Institute for Agriculture, Environment and Development, Departamento de Fitotecnia, Escola de Ciências e Tecnologia, Universidade de Évora, 7006-554 Évora, Portugal
[2] MED—Mediterranean Institute for Agriculture, Environment and Development, Departamento de Química e Bioquímica, Escola de Ciências e Tecnologia, Universidade de Évora, 7000-671 Évora, Portugal
[3] Division of Horticultural Sciences, University of Bonn, Auf dem Hügel 6, 53121 Bonn, Germany
* Correspondence: rmam@uevora.pt (R.M.A.M.); raf@uevora.pt (R.F.)

**Abstract:** The purpose of this study was to develop substrates with little or no peat by combining coir-based growing media with municipal compost and/or acacia biochar, two locally produced renewable resources, and to assess their effects on lettuce seedling emergence and growth, as well as their content in photosynthetic pigments and total phenols. Two experiments were carried out, the first with six mixes using compost and biochar blended with perlite, pine bark, and blonde peat to adjust some physicochemical characteristics. The mixes of coir: compost: pine bark: blonde peat (73:12:5:10, *v/v*) and coir: compost: biochar: blonde peat (73:12:10:5, *v/v*) had physicochemical characteristics closer to or within the normal range of the substrates. The presence of 12% compost and 10% biochar in the mixtures had no adverse effect on lettuce seed germination and cumulative seed emergence, which ranged from 90 to 99%. The seedling growth in those mixes was vigorous and higher than in other mixtures. Coir-based growing media with municipal solid waste compost and compost plus biochar can reduce the use of peat to a percentage of 5–10% *v/v* and the use of 17–22% *v/v* of locally produced renewable resources. In addition, mixtures affected the total phenol content in the lettuce leaves. Future research is needed to assess the behavior of seedlings after their transplantation.

**Keywords:** *Lactuca sativa* L.; sustainable substrate; peat alternatives; pH; electrical conductivity; seedling emergence; total phenols





## 1. Introduction

Transplanting seedlings is the most-common method for vegetable crop establishment in Portugal and it is increasing worldwide. It promotes plant growth, early maturation, yield, harvest-time plant uniformity, and efficient land use. Furthermore, it may require less irrigation water and herbicides than crops established by sowing [1].

Peat or peat-rich substrates are the most-common growing medium for vegetable transplants. However, peat is a non-renewable resource and its extraction has detrimental effects on the environment and ecosystems. In addition, peatlands are natural carbon sinks [2,3]. Hence, when peat is used as a substrate, stored carbon is released, negatively affecting $CO_2$ balance [3]. Therefore, peat use in horticulture is restricted or regulated, mainly in some European countries [4,5]. As a result, using less peat is the primary goal of soilless-grown plants today [2,5,6].

An alternative strategy could be to use selectively collected municipal solid organic waste (MSW) and biochar, two organic resources, in substrates [7–9]. The use of MSW contributes to reducing organic waste accumulation in landfills and carbon footprint while

also boosting nutrient recycling [10–12]. Perhaps the most beneficial effect of compost inclusion in a growth medium is its nutritional contribution. Matured compost may act as a slow-release fertilizer [13]. Additionally, it might have a biostimulant effect and suppress plant diseases caused by various pathogens and pests [1,14]. Organic compost tends to have peat-like porosity and aeration properties [15].

Biochar can be a beneficial component in substrates as it can increase plant growth, reduce dependence on non-renewable substrate components, and contribute to carbon sequestration [7]. In addition, biochar can replace peat [16,17], perlite, and vermiculite without compromising seedling quality [17].

Raw materials used as growing media constituents should be free from phytotoxic compounds [18] and should demonstrate good chemical properties, such as a suitable pH [19,20] and the content of certain elements and/or salt content [5,21–23]. However, municipal organic compost and biochar have some chemical characteristics, such as pH and electrical conductivity (EC), that may negatively affect plant growth, limiting their use as stand-alone substrates. For example, most biochar often has a high pH [20,24]. The pH of biochar varies depending on the feedstock type, temperature during its production [25,26], and, as recently investigated, particle size. The fractions from the same biochar can have different pH levels [5,20]. In addition, MSW compost usually has high electrical conductivity and pH values [27].

Finally, it should be noted that plants are more sensitive to salt stress at earlier plant-growth stages (germination, seedling, establishment) than plants at later stages [28]. Therefore, organic composts may also have human pathogens. To reduce them in the future, the compost must be certified as to the raw materials used and the maximum temperature and time of exposure to these during the thermophilic phase. The use of green compost can reduce the risk of human pathogens' presence [4]. A mixture of MSW and coir could overcome its limitations. Coir has a low pH and density, good physical stability, aeration, and water-holding capacity [9,29–31]. The effects of biochar incorporation on plant growth in container substrates depend on biochar properties, plant type, percentage of biochar applied, and other container substrate components mixed with biochar [32]. For example, tomato plant heights and bell pepper (*Capsicum annuum* L.) dry weights increased with the addition of 1, 3, and 5% ($w/w$) to a soilless mixture of coconut fiber and tuff (volcanic ash) [33]. On the other hand, the mix of coir with MSW and biochar in a ratio of 3:1 (75% coir by volume in the mixture) decreased the EC and pH, but not to adequate levels [34].

Municipal compost and biochar are two locally produced renewable resources whose use lessens Portugal's dependency on peat and coir imports, keeps organic waste out of landfills, and reduces the carbon footprint [35] and greenhouse gas (GHG) emissions [12].

This research aimed to reduce the use of peat in substrates by investigating the suitability of coir-based substrates in combination with selectively harvested municipal solid organic waste and/or acacia biochar for successful horticultural plant cultivation. Further, we investigated the effect of mixes on lettuce seedling emergence, growth, and the content of photosynthetic pigments and total phenols.

## 2. Materials and Methods

Two experiments were conducted at the Center of Studies and Experimentation of Mitra in Évora, Portugal (38°57′ N. 8°32′ W. elevation 200 m).

We designed the first experiment to create mixes with the characteristics of an ideal substrate. First, we mixed a coir-based medium with municipal compost or biochar to achieve adequate substrate characteristics. Next, we added perlite, blonde peat, and pine bark to blend the mixes.

Further, we evaluated how adding fertilizer to mixes affected their physicochemical properties. Finally, we investigated the emergence and growth of lettuce seedlings and the levels of photosynthetic pigments and total phenols in the leaves.

The second experiment was realized in sequence with the first. The goal was to enhance the mixes' physicochemical properties, particularly pH and EC. Moreover, we

aimed to evaluate these effects on seedling growth, photosynthetic pigment content, and total phenol content in a lettuce.

*2.1. Components of Mixes*

The following components were used to make the mixtures: coir, municipal solid organic compost, acacia wood biochar, perlite, pine bark, and blonde peat.

A 100% coir pith was used. The coir had a pH of 5.5 to 6.0, an electrical conductivity (EC) greater than 1.5 dS m$^{-1}$, granulometry 0–10 mm, total porosity = 95% *v/v*, air = 25% *v/v*, and CEC = 60–120 meq/100 g. The physiochemical characteristics of compost (Nutrimais, Lipor Company, Baguim do Monte, Portugal) and acacia wood biochar (Ibero Massa, Oliveira de Azeméis, Portugal) are presented in an earlier study [35]. Biochar was pyrolyzed at a temperature of 400 to 500 °C. The raw materials used in the "Nutrimais" manufacturing process include horticultural products, food scraps carefully selected from restaurants, canteens, and similar establishments, forest exploitation residues (e.g., branches and foliage), and green residues (e.g., flowers, grasses, and pruning). According to the manufacturer, the compost used in this study is free of pathogens. The compost had wood fragments larger than 1 cm, which were ground down to a particle size of between 3 and 4 mm to allow for the introduction of the mixtures into the wells of the plastic trays.

Perlite (Knauf, Dortmund, Germany) has particles from 2 to 6 mm (coarse perlite), is pH-neutral, and is chemically inert. The pH and the electrical conductivity (1:5 H$_2$O) of the blonde peat (Greenterra Ltd., Riga, Latvia) were, respectively, 5.5 to 6.5 and 1 dS m$^{-1}$. The pH and the EC of the components in the mixes were measured in the aqueous extract (1:5 substrate: distilled water, *w/v*) according to the methodology presented by Machado et al. (2021). The nitrate (NO$_3$-N) levels in aqueous extracts (1:5 substrate:water, *v/v*) were also determined using an ion-specific electrode and meter (Crison Instruments, Barcelona, Spain).

*2.2. Seedling-Growth Experiments*

2.2.1. Growth Conditions and Mixes

The experiments were conducted between 27 March and 11 May 2022. The average daily temperature inside the greenhouse at the shoot seedling level ranged from 17 to 26 °C. These values were within the range of temperatures suitable for the germination of lettuce seeds (15 to 25 °C) [36].

Solar radiation ranged from 127.3 to 348.8 W·m$^{-2}$·d$^{-1}$. Seeds of lettuce (*Lactuca sativa* L. cv. Grand Rapids) with a mean germination rate of 95%, evaluated through a germination test, were used in both experiments.

Experiment 1 was carried out with twelve treatments: six mixes unfertilized and fertilized (Table 1).

**Table 1.** Constitution and proportion of the different components in the mixes (experiment one).

| Mixes [1] (Treatments) | Mixes (%, *v/v*) | | | | | |
|---|---|---|---|---|---|---|
| | C | B | MSW | P | Pi | BP |
| C + B + P | 84 | 14 | - | 2 | - | - |
| C + B + Pi | 70 | 20 | - | - | 10 | - |
| C + B + Pi + BP | 65 | 20 | - | - | 5 | 10 |
| C + MSW + P | 84 | - | 14 | 2 | - | - |
| C + MSW + Pi | 70 | - | 20 | - | 10 | - |
| C + MSW + Pi + BP | 65 | - | 20 | - | 5 | 10 |

[1]—C—Coir, B—Biochar, MSW—Municipal solid organic waste, P—Perlite, Pi—Pine bark, BP—Blonde peat.

The mixes were fertilized with a 1 g controlled-release fertilizer (6N-5.3P-10K + 1.2% Mg + 0.02% B + 0.05% Cu, 0.2% Fe, 0.06% Mn 0.02%, Mn and 0.015% Zn) l$^{-1}$ of growing media. Each replicate's treatments (mixes) occupied two rows of plastic trays (26 wells). Each well had a volume of 25 cm$^3$.

The second experiment was carried out with five mixes, whose constitution is presented in Table 2. At each mix, we added 1 g of controlled-release fertilizer (6N-5.3P-10K + 1.2% Mg + 0.02% B + 0.05% Cu, 0.2% Fe, 0.06% Mn, 0.02% Mn, and 0.015% Zn) per L of growing medium.

**Table 2.** Constitution and proportion of the different components in the mixes (experiment two).

| Mixes [1] (Treatments) | Mixes (%, *v/v*) | | | | | |
| | C | MSW | B | P | Pi | BP |
|---|---|---|---|---|---|---|
| C + MSW + P | 85 | 13 | - | 2 | - | - |
| C + MSW + BP | 80 | 12 | - | - | - | 8 |
| C + MSW + Pi | 80 | 12 | - | - | 8 | - |
| C + MSW + Pi + BP | 73 | 12 | - | - | 5 | 10 |
| C + MSW + B + BP | 73 | 12 | 10 | - | - | 5 |

[1]—C—Coir, B—Biochar, MSW—Municipal solid organic waste, P—Perlite, Pi—Pine bark, BP—Blonde peat.

Both experiments were arranged in a complete randomized block design with five replicates. Each replicate's treatments (mixes) occupied two rows of plastic trays (26 wells). Each well had a volume of 25 cm$^3$. The seeds were manually sown in plastic trays; one seed was placed in each compartment at 1.5 cm depth and covered. Nursery trays were watered by micro sprinklers three to six times per day in order to keep the substrate well-moistened.

Fresh tap water was used for irrigation; it had an electrical conductivity (EC) of 0.3 d$S$ m$^{-1}$, a pH of 7, and 0.10 to 0.30 mmol L$^{-1}$ NO$_3$, 0.12 to 0.20 mmol L$^{-1}$ Ca, 0.15 to 0.22 mmol L$^{-1}$ Mg, and 2.1 mmol L$^{-1}$ Cl and 0.7 mmol L$^{-1}$ Na.

### 2.2.2. Measurements

In both experiments, the same methodology was used to evaluate the initial physicochemical properties of the mixes and their effects on the emergence and growth of seedlings. Thus, the measurements made will be presented together. The initial physicochemical characteristics of the mixtures measured were pH, EC, mass wetness, moisture content, total porosity, and bulk density. The pH and the EC were measured in the aqueous extract (1:5 substrate: water, *w/v*) according to the methodology presented by [9]. Moisture content, total porosity, and bulk density were determined following the methodology described in [37]. The number of seedlings that emerged in each mix of all replicates was recorded throughout the experimental period. In each treatment, six seedlings were randomly collected from each replication. In these, the weight of the root system and the shoot, the number of leaves, and the leaf area were measured. The root system of the seedlings was separated from the substrate by washing it in running water with a net underneath to avoid root loss. We measured the leaf area using a leaf area meter (LI-COR Model LI–3000A).

Leaf samples of 0.5000 g from three lettuce plants were collected from all repetitions in each treatment. The seedlings were macerated in a mortar and homogenized in 4 mL of methanol/water (90:10, *v/v*; MW90 extract) or methanol/water (80:20, *v/v*; MW80 extract) for 1 min. Aliquots of the methanolic extracts MW90 or MW80 were obtained after centrifugation at 4 °C and 6440× *g* for 5 min was preserved at −20 °C for later use [37].

The following equations were used to determine the concentration (mg/100 g FW) of chlorophyll a (Chl a), chlorophyll b (Chl b), and carotenoids (Cc) of MW90 extract, where A denotes absorbance, following [38]:

1.  Chl a = 16.82A$_{665.2}$ − 9.28A$_{652.4}$
2.  Chl b = 36.92A$_{652.4}$ − 16.54A$_{665.2}$
3.  Cc = (1000A$_{470}$ − 1.91Chl-a − 95.15Chl-b)/225

The total phenolic compound (TPC) content in the MW80 extract was determined according to that described by [39] by reacting an appropriate volume of sample or standard with 1/10 diluted Folin–Ciocalteau reagent and 7.5% sodium carbonate. After stirring the reaction mixture in the vortex, we waited for 90 min at room temperature in the dark. The absorbance

of the chromophore then formed and was read at 760 nm. Finally, the TPC concentration, expressed as milligrams of gallic acid equivalent (GAE) per 100 g of fresh weight (FW), was calculated using a calibration curve (GAE, *n* = 6 concentrations from 0 to 200 mg/L).

### 2.2.3. Data Analysis

Data were analyzed using analysis of variance using SPSS 25 software (Chicago, IL, USA) licensed to the University of Évora. The means were separated at the 5% level using Duncan's new multiple-range test.

## 3. Results and Discussion

### *3.1. Physicochemical Characteristics of the Components*

Table 3 shows some of the characteristics of the components used in the mixes. The pH and EC were relatively high in the municipal solid organic compost, as was the nitrate content. Further, also noteworthy was the high pH of biochar (8.76), while the EC was low (0.22 dS m$^{-1}$) (Table 3).

**Table 3.** Physicochemical characteristics of the components.

| Components | pH | EC (dS m$^{-1}$) | Bulk Density (g cm$^{-3}$) | Nitrate (NO$_3^-$) (ppm) |
|---|---|---|---|---|
| Coir | 5.66 | 1.5 | 0.12 | - |
| MSW [1] | 7.91 | 8.62 | 0.23 | 91.1 |
| Biochar [2] | 8.76 | 0.22 | 0.36 | 4.45 |
| Pine bark | 4.84 | 0.13 | 0.18 | 12.1 |
| Perlite | 7.06 | 0.04 | 0.14 | - |
| Blonde peat | 5.5 | 0.11 | 0.12 | - |

[1]—MSW—Municipal solid organic waste. [2]—The granulometry of the biochar was also determined through the use of sieves, as described by [40]. The biochar granulometry, expressed as a percentage by weight, was: $\geq$ 2 mm (28.11%); $\geq$1 mm < 2 mm (30.05%); $\geq$ 0.5 mm < 1 mm (15.60%); < 0.5 mm (26.24%).

### *3.2. Experiment 1*

#### 3.2.1. Initial Physicochemical Characteristics of the Mixes

The physicochemical properties of the mixes were unaffected by the interactions of treatments with the fertilizer supply (Table 4). Despite the initial pH of biochar being higher than that of MSW (Table 1), the pH of mixtures containing biochar ranged from 7.14 to 7.77, while that of mixtures including compost ranged from 7.81 to 8.09. This was probably due to the high cation-exchange capacity of composts, which increased the buffering capacity of the growing medium [41]. On the other hand, fresh biochar typically has a low CEC, as the high temperatures during pyrolysis reduce the concentration of functional groups (e.g., –OH, –COOH, –CH, and –C=O) [42].

**Table 4.** Physicochemical characteristics of the mixes of experiment 1.

| Mixes [1] | pH | EC (dS m$^{-1}$) | Bulk Density (g cm$^{-3}$) | Mass Wetness [3] (g Water/g Substrate) | Total Porosity (%) | Moisture Content (%, *w/w*) |
|---|---|---|---|---|---|---|
| C + B + P | 7.51 c [2] | 1.60 c | 0.18 bc | 5.39 ab | 97.95 a | 81.06 ab |
| C + B + Pi | 7.77 b | 1.14 d | 0.18 c | 4.96 bc | 98.75 a | 79.56 bc |
| C + B + Pi + BP | 7.14 d | 0.98 d | 0.18 c | 5.03 bc | 99.22 a | 78.17 c |
| C + MSW + P | 7.81 b | 2.80 b | 0.18 c | 5.71 a | 98.90 a | 82.88 a |
| C + MSW + Pi | 8.09 a | 3.42 a | 0.21 a | 4.92 bc | 99.28 a | 81.73 ab |
| C + MSW + Pi + BP | 7.95 ab | 3.25 a | 0.19 b | 4.69 c | 98.66 a | 82.11 ab |
| Significance | *** | *** | *** | *** | NS | *** |

[1]—C—Coir, B—Biochar, MSW—Municipal solid organic waste, P—Perlite, Pi—Pine bark, BP—Blonde peat. [2]—Means followed by different letters within a column are significantly different. *** significant at $p < 0.001$ level. NS—not significant. Mean separation was performed using Duncan's multiple-range test. Means are based on four replicates. [3]—Mass wetness—the water content of a sample on a dry mass basis; this is calculated as (wet weight—dry weight)/dry weight.

Regardless of the differences, the pH values of the different blends were higher than the maximum value of the adequate range for plant growth in substrates (6.5) [43–45]. As a result, the ratios of the components in the mixes of experiment 2 (Table 2) were altered to decrease the pH.

The EC of mixes with biochar (ranging from 0.98 to 1.60 dS m$^{-1}$) was much lower than that of mixtures with MSW (Table 4). Regarding the mixes with biochar, compost led to increases in EC of 1.2 to 2.3 dS m$^{-1}$. EC values in mixes with compost ranged from 2.80 to 3.42 dS m$^{-1}$ (Table 3), which may influence seed germination and seedling growth. Lettuce is moderately sensitive to salinity, having a salinity threshold of 2 dS m$^{-1}$ in soil. Nevertheless, plants in their early stages (germination, seedling) are more susceptible to salt stress than plants in their later stages [28]. Although the EC value of substrates varies depending on the method used to determine it [46,47], the highest EC level in the range appropriate for growing plants in substrates is generally higher than in soil. According to Martinez and Roca [44], the appropriate range of the EC for substrates ranges from 0.75 to 3.5 dS m$^{-1}$, but they did not discriminate the method used to determine EC. Salinity levels ranging from 2 to 3.49 dS m$^{-1}$ in saturated media extract are satisfactory for most plants, but the growth of some sensitive plants may be reduced [46]. The EC in mixes with MSW was lower in the mixture (coir + MSW + perlite) (2.8 dS m$^{-1}$) than in the other mixes. As a result, the change in the proportions of the components in this mixture in experiment 2was reduced (Table 2).

Bulk density ranged from 0.18 to 0.21 g cm$^{-3}$. These values were adequate for substrates [45,48,49]. However, the bulk density of an ideal substrate for vegetable seedlings should not exceed 0.4 g cm$^{-3}$ [43].

Although perlite is generally added to the substrate to increase the proportion of large pores and reduce the water-holding capacity, the mass wetness was higher in mixes with perlite. This may be due to the low proportion of perlite added (2%) (Table 1). All mixtures had a total porosity above 85% (from 97.9 to 99.3%), which is regarded as suitable for substrates [46] (Table 4). Moisture content ranged from 78.2 to 82.8, with the coir + biochar + perlite + biochar mix having a lower moisture content.

### 3.2.2. Seed Emergence

The addition of fertilizer and the interaction between treatments did not affect the seedling emergence. At 5 DAS (days after sowing), the percentage of seed emergence was affected by the mixture (Figure 1). The seed emergence was higher in mixes with biochar at 5 DAS, ranging from 97 to 100%. Low rates of biochar can have a stimulatory effect on germination [42]. The seed emergence precocity was lower in the coir + MSW + blonde peat + pine bark (65:20:5:10. *v/v*) and coir + MSW + pine bark (70:20:10, *v/v*) mixes as compared to other mixtures. This may be due to the high percentage of MSW in the mixture (20%, *v/v*) (Table 1). Reference [50] reported that the percentage of MSW in mixes with peat affected seed emergence. However, at 16 DAS, the cumulative seedling emergence ranged from 98 to 100% and was not significantly affected by the mixes ($p < 0.05$). This indicates that the presence of MSW and biochar in percentages ranging from 14 to 20% *v/v* did not affect the germination since the average cumulative seedling emergence and seedling survival were higher than the average germination rate of the seeds determined (95%).

### 3.2.3. Photosynthetic Pigments and Total Phenols

The interaction between treatments significantly affected leaf photosynthetic pigments and total phenol content (Table 5). However, adding fertilizer appears to increase the content of chl a, chl b, total chl, and carotenoids (Cc) in all substrates. Nutrient availability is essential for photosynthetic pigment biosynthesis [51]

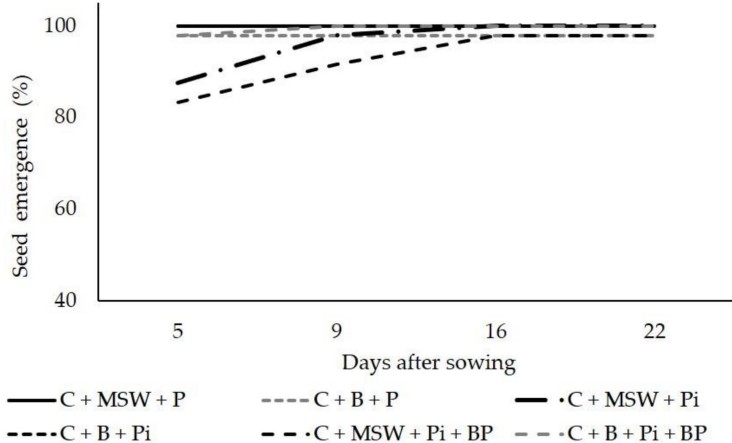

**Figure 1.** Influence of mixes on cumulative seedling emergence (C—Coir, B—Biochar, MSW—Municipal solid organic waste, P—Perlite, Pi—Pine bark, BP—Blonde peat).

**Table 5.** Effect of fertilization and mixes on leaf photosynthetic pigments and in total phenol content.

| Mixes [1] | Chl a | Chl b | Chl Total | Cc | TPC |
|---|---|---|---|---|---|
| | (mg 100 g$^{-1}$ FW) | | | | (mg GAE 100 g$^{-1}$ FW) |
| Unfertilized | | | | | |
| C + B + P | 7.90 def [2] | 9.69 d | 17.59 e | 5.22 d | 68.16 fg |
| C + B + Pi | 8.49 cde | 8.82 d | 17.32 e | 6.98 c | 149.37 a |
| C + B + Pi + BP | 7.64 def | 9.50 d | 17.14 | 7.82 c | 122.70 b |
| C + MSW + P | 6.37 fg | 9.57 d | 15.94 ef | 7.13 c | 104.05 cd |
| C + MSW + Pi | 7.40 efg | 8.34 d | 15.74 ef | 5.50 d | 67.58 fg |
| C + MSW + Pi + BP | 5.83 g | 7.98 d | 13.81 f | 4.84 d | 115.88 bc |
| Fertilized | | | | | |
| C + B + P | 14.73 a | 19.18 a | 33.91 a | 10.40 b | 68.27 fg |
| C + B + Pi | 12.80 b | 17.9 ab | 30.71 b | 12.88 a | 62.67 fg |
| C + B + Pi + BP | 9.28 cd | 12.06 c | 21.34 d | 12.80 a | 107.01 bcd |
| C + MSW + P | 9.3 cd | 13.49 c | 22.80 d | 9.18 b | 77.26 ef |
| C + MSW + Pi | 9.64 c | 16.13 b | 25.77 c | 12.72 a | 54.43 g |
| C + MSW + Pi + BP | 10.16 c | 16.29 b | 26.44 c | 13.40 a | 93.42 de |
| Significance | | | | | |
| Fertilizer | *** | *** | *** | *** | *** |
| Mixes | *** | *** | *** | *** | *** |
| Interaction | *** | *** | *** | *** | *** |

[1]—C—Coir, B—Biochar, MSW—Municipal solid organic waste, P—Perlite, Pi—Pine bark, BP—Blonde peat. FW—Fresh weight. [2]—Means followed by different letters within a column are significantly different. *** significant at $p < 0.001$ level. NS—not significant. Mean separation was performed using Duncan's multiple-range test.

Chl a, Chl b, and total Chl contents were higher in the fertilized coir + biochar + perlite mix (Table 5). Chl a, Chl b, and total Chl contents ranged from 5.8 to 14.7, 7.98 to 19.2 and 13.8 to 33.9 mg/g of leaf fresh weight, respectively. These values were in the same range or slightly higher than those reported by [51] for lettuce seedlings. Chl b, as reported by [52], also had higher contents than Chl a.

Regarding the total content of phenols, it appears that adding fertilizer contributed to its decrease, except in the coir + biochar + perlite mix. This indicates that the seedlings may have been subjected to significant abiotic stress in the unfertilized mixtures, probably due to nutrient deficiency. Nutrient deficiency in lettuces increases total phenol content [53].

The fertilized mixes with four components, with blonde peat (10%, *v/v*), had the highest levels of total phenols (Table 5). The TPC in the different treatments ranged from 54.43 to 149.37 (mg GAE 100$^{-1}$ FW). These values are lower than those mentioned in lettuce seedlings by [23], which range between 400 and 600 mg GAE 100$^{-1}$ FW. However, as is

known, TFC is affected by many factors, including genotype, growing conditions, and others [54].

### 3.2.4. Seedling Growth

Growth parameters, except for dry matter %, were not significantly affected by the interaction of treatments (Table 6).

**Table 6.** Effect of the mix on seedling growth, experiment 1.

| Mixes | Shoot Fresh Weight | Shoot Dry Weight | Seedling Total Dry Weight | Seedling Dry Weight | Leaf Area | Leaves |
|---|---|---|---|---|---|---|
| | (g/Plant) | | | (%) | (cm$^2$) | (N°) |
| Unfertilized | | | | | | |
| C + B + P [1] | 0.89 def [2] | 0.09 ef | 0.14 e | 8.21 a | 21.6 def | 6.00 fg |
| C + B + Pi | 0.67 f | 0.06 f | 0.09 f | 6.24 b | 16.75 f | 5.92 fg |
| C + B + T + Pi | 0.72 ef | 0.07 f | 0.09 f | 6.40 b | 19.02 ef | 5.58 g |
| C + MSW + P | 1.08 cde | 0.10 de | 0.14 de | 6.03 b | 28.37 cde | 6.67 def |
| C + MSW + Pi | 1.73 b | 0.14 c | 0.20 c | 5.94 b | 46.33 b | 7.08 cde |
| C + MSW + Pi + BP | 2.05 b | 0.18 b | 0.25 b | 6.44 b | 54.02 ab | 7.75 bc |
| Fertilized | | | | | | |
| C + B + P | 1.19 cd | 0.12 cde | 0.18 cd | 6.44 b | 31.08 cd | 7.58 bcd |
| C + B + Pi | 1.21 cd | 0.12 cde | 0.17 cde | 6.84 b | 31.28 cd | 7.33 cde |
| C + B + Pi + BP | 1.22 cd | 0.13 cd | 0.18 cde | 6.50 b | 31.88 cd | 6.50 efg |
| C + MSW + P | 1.35 c | 0.13 cd | 0.19 c | 6.48 b | 35.62 c | 7.33 cde |
| C + MSW + Pi | 1.84 b | 0.17 b | 0.23 b | 6.13 b | 49.90 b | 8.50 b |
| C + MSW + Pi + BP | 2.57 a | 0.22 a | 0.29 a | 5.97 b | 66.11 a | 9.50 a |
| Significance | | | | | | |
| Fertilizer | *** | *** | *** | NS | *** | *** |
| Mixe | *** | *** | *** | * | *** | *** |
| Interaction | NS | NS | NS | * | NS | NS |

[1]—C—Coir, B—Biochar, MSW—Municipal solid organic waste, P—Perlite, Pi—Pine bark, BP—Blonde peat. [2]—Means followed by different letters within a column are significantly different. * and *** significant at $p < 0.05$ and 0.001 levels, respectively. NS—not significant. Mean separation was performed using Duncan's multiple-range test.

Lettuce seedling growth was significantly affected by fertilizer addition and growing media mixtures (Table 6). In unfertilized mixes, seedlings grown in biochar grew less than those grown in mixes with MSW. This indicates that MSW contributed to seedling nutrition and that mixes with biochar have a lower ability to feed them. The EC of mixes with biochar had low EC values (0.98 to 1.60 dS m$^{-1}$). Biochar had a lower EC and nitrate level than MSW (Table 3). Biochar has a low content of extractable macronutrients, except for K [55]. Chrysargyris et al. [24] also reported that applying fertilizer to mixtures containing biochar can increase the growth of lettuce seedlings, but it depends on the percentage of biochar in the mix.

On the other hand, the biochar particle size of the fractions from the same biochar could also influence pH and the nutrient availability of Ca and Mg. This could lead to nutrient imbalances during the cultivation of plants [5,20]. Thus, future research is required to determine whether the lower growth of seedlings on substrates containing biochar is due to a nutritional deficit.

Additional fertilizer to the mixes with biochar increased seedling growth (Table 6). Fertilizer addition to mixes with MSW also increases the shoot and total dry weight. The total dry weight of the seedlings in mixtures containing the same proportion of MSW and biochar (Table 1), without or with fertilizer, was significantly higher in the mixtures containing MSW.

Seedling growth in coir + MSW + pine bark + blonde peat (65:20:5:10, *v/v*) and coir + MSW + pine bark (70:20:10, *v/v*) mixes was higher than that in the other mixes (Table 6). Compared to the Coir + MSW + pine bark mix, the mix with 10% blond peat (Coir + MSW +

pine bark + blonde peat) boosted shoot dry and total dry weight by nearly 30%. The seedlings grown in these mixes, despite having a pH > 7.9 and an EC > 3.2 (Table 3), presented a higher growth than those grown in the other mixes. They did not present any visual symptoms of nutrient deficiencies or excess salts and the roots were healthy (Figure 2). This may be due to the presence of humic acids, which account for 13% [34] of the dry weight of the compost, which was higher in these mixes (20%). Humic acids may contribute to the availability of nutrients, especially micronutrients, by chelating and co-transporting micronutrients to plants [56] and increasing $H^+$ exudation [57].

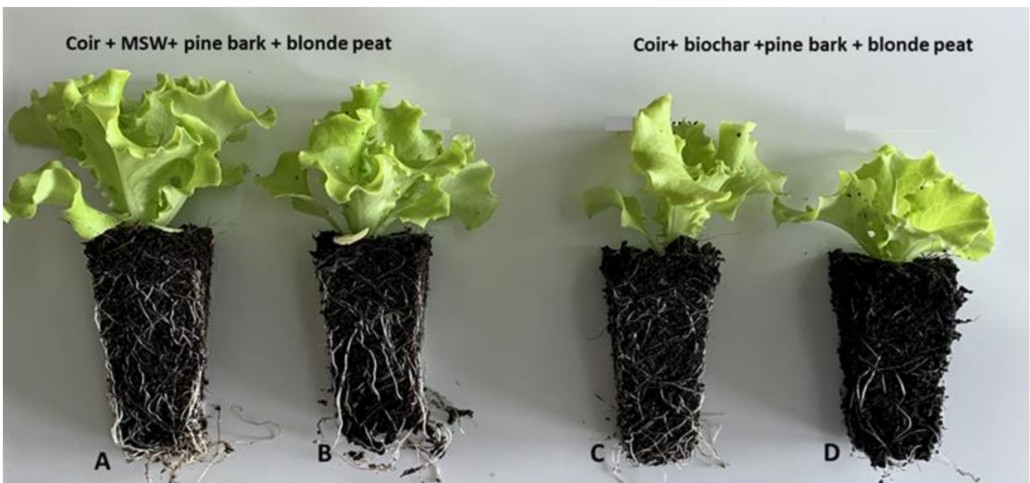

**Figure 2.** Seedlings grown in coir + MSW + pine bark + blonde peat (65:20:5:10, $v/v$) (**A**—fertilized, **B**—unfertilized) and in coir + biochar + pine bark + blonde peat (65:20:5:10, $v/v$) (**C**—fertilized, **D**—unfertilized).

Although plants are more sensitive in the initial phase, lettuce seedlings from the mixes with high EC (> 3.2) did not present any visual symptoms of excess salts. According to [58], the initial EC of the mixes with compost assessed in the saturated extract should not exceed 2.5 d$S$ m$^{-1}$ for tomato seedlings, which are more tolerant to salt stress than lettuce. However, the response to salinity depends on environmental conditions and the moisture content of the substrate. In the present study, the effects of salt stress may have been reduced due to mild temperatures and frequent irrigation that decrease the substrate's osmotic potential. On the other hand, humic acids may also reduce the salt stress effects since they may increase osmoprotection and ion homeostasis [59,60].

### 3.3. Experiment 2

3.3.1. Initial Physicochemical Characteristics of the Mixes

The average pH values were affected by the mix. The mixes with blonde peat had a lower pH than other mixes (Table 7). In the mixes' coir + organic compost + blonde peat and coir + organic compost + pine bark + blonde peat, the pH (6.56) was within the range considered suitable for substrates. In the mixture with biochar (10%. $v/v$), the average pH value (7.16) was slightly higher than the maximum value of the adequate range. Except for the coir + MSW + perlite mix, the goal of lowering the initial pH in the mixes with MSW was met. The mixes also affected EC, ranging from 2.44 to 2.79 dS/m (Table 7). Despite the differences in EC of the mixes, they are within an adequate range for substrates, as previously mentioned. The addition of blonde peat to the mixtures contributed to the decrease in EC (Table 7).

Bulk density ranged from 0.12 to 0.14 g/cm$^3$ (Table 7). As previously mentioned, these values were within an adequate range for substrates. The total porosity was not significantly affected by the mixtures and was above 85%, as required for substrates. Mass wetness was consistently higher than 6.32 g of water per g substrate in all growing media, and their values increased relative to the previous experiment.

**Table 7.** Physicochemical characteristics of the mixes of experiment 2.

| Mixes [1] | pH | EC (dS m$^{-1}$) | Bulk Density (g/cm$^3$) | Mass Wetness [3] (g Water/g Substrate) | Total Porosity (%) | Moisture (% *w/w*) |
|---|---|---|---|---|---|---|
| C + MSW + P | 7.25 a [2] | 2.79 a | 0.12 c | 7.38 a | 98.58 a | 75.03 b |
| C + MSW + BP | 6.56 b | 2.56 bc | 0.12 c | 7.18 a | 98.67 a | 75.12 b |
| C + MSW + Pi | 7.17 a | 2.73 b | 0.13 b | 6.82 b | 98.52 a | 77.64 a |
| C + MSW + Pi + BP | 6.56 b | 2.44 c | 0.13 b | 6.67 b | 98.62 a | 74.74 b |
| C + MSW + B + BP | 7.16 a | 2.53 bc | 0.14 a | 6.32 c | 98.45 a | 76.23 ab |
| Significance | *** | *** | *** | *** | NS | * |

[1]—C—Coir, B—Biochar, MSW- Municipal solid organic waste, P—Perlite, Pi—Pine bark, BP- Blonde peat. [2]—Means followed by different letters within a column are significantly different. * and *** significant at $p < 0.05$ and 0.001 levels, respectively. NS—not significant. Mean separation was performed using Duncan's multiple-range test. Means are based on four replicates. [3]—Mass wetness—the water content of a sample on a dry mass basis; this is calculated as (wet weight—dry weight)/dry weight.

### 3.3.2. Seed Emergence

Seedling emergence in coir + compost + blond peat was lower than in the other mixes but still very high (91%) (Figure 3). At 9 DAS, the emergence was rapid in the remaining substrates, ranging from 91 to 100%. In these mixes at 23 DAS, cumulative seedling emergence ranged from 97 to 100%. The presence of MSW (12–13%) and MSW (12%) + biochar (10%) in the mixture, as in the previous experiment, had no significant effect on seed germination and cumulative seed emergence.

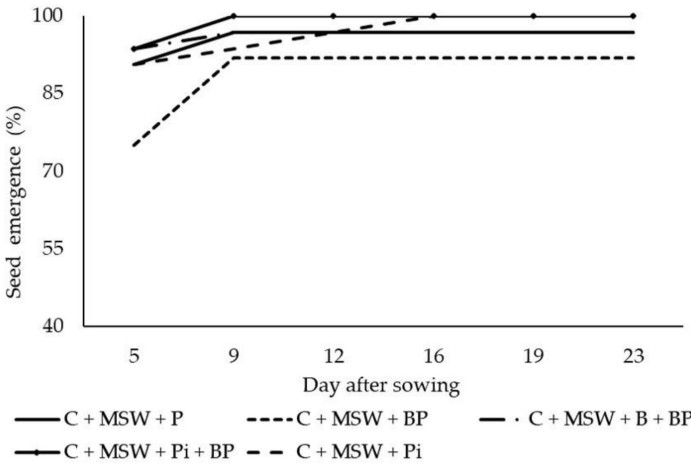

**Figure 3.** Influence of mixes on cumulative seedling emergence (C—Coir, B—Biochar, MSW—Municipal solid organic waste, P—Perlite, Pi—Pine bark, BP—Blonde peat).

### 3.3.3. Photosynthetic Pigments and Total Phenols

The mixes affected the average content of photosynthetic pigments in the leaves. For example, seedlings grown in the coir + MSW + blonde peat (80:12:8. *v/v*) mix had higher levels of chl a, chl b, total chl, and carotenoids in their leaves than plants grown in the other mixes (Table 8).

Chl a, chl b, and total Chl content ranged from 10.1 to 13.2, 13.5 to 15.4, and 24.8 to 29.7 mg/g of leaf fresh-weight, respectively. These values were slightly higher than those reported by [52] for lettuce seedlings.

The average TPC in the different mixes ranged from 45.79 to 70.04 mg GAE 100$^{-1}$ FW (Table 8). These values were lower than those reported by [24] for lettuce seedlings.

The TPC of seedlings grown in the mixes coir + MSW + biochar + blonde peat (45.79 mg GAE 100 g$^{-1}$ FW) and Coir + MSW + pine bark (49.05 mg GAE 100 g$^{-1}$ FW) was lower than that grown in the other mixes. In lettuce seedlings grown on substrates with blonde peat and biochar, the total phenol content decreased with the addition of biochar, regardless of fertilization [24].

The TPC in the seedlings of the other mixes was higher than in previous mixes. Seedlings with high total phenol content may have a more remarkable ability to resist abiotic stress after transplantation. Thus, future research is required to assess how the seedlings from different mixes behave following transplantation.

**Table 8.** Effect of mix on leaf photosynthetic pigments and in total phenol content.

| Mixes [1] | Chl a | Chl b | Chl Total | Cc | TPC |
|---|---|---|---|---|---|
| | (mg.100 g$^{-1}$ FW) | | | | (mg GAE 100 g$^{-1}$ FW) |
| C + MSW + P | 11.92 ab [2] | 15.54 ab | 27.46 ab | 13.37 b | 69.63 a |
| C + MSW + BP | 13.15 a | 16.50 a | 29.65 a | 15.28 a | 62.20 a |
| C + MSW + Pi | 11.79 ab | 15.45 ab | 27.23 ab | 11.60 c | 49.05 b |
| C + MSW + Pi + BP | 10.10 b | 13.85 b | 24.84 b | 9.80 d | 70.04 a |
| C + MSW + B + BP | 12.55 ab | 13.53 b | 26.08 ab | 10.42 d | 45.79 b |
| Significance | ** | ** | ** | ** | * |

[1]—C—Coir, B—Biochar, MSW—Municipal solid organic waste, P—Perlite, Pi—Pine bark, BP—Blonde peat, FW—fresh weight. [2]—Means followed by different letters within a column are significantly different. * and ** significant at $p < 0.05$ and 0.01 levels, respectively. NS—not significant. Mean separation was performed using Duncan's multiple-range test.

### 3.3.4. Seedling Growth

The mixes significantly affected seedling growth, which was higher in the mixtures with four components than in the mixtures with three (Table 9). Seedling shoot (0.20 g/plant) and total dry weight (0.27 g/plant) were higher in seedlings grown in the coir + MSW + pine bark + blonde peat (73:12:5:10, *v/v*) mix than the other mixes. It should be noted, nonetheless, that seedling shoot fresh weight, leaf area, number of leaves, and total dry weight, grown in the mix with biochar coir + MSW + biochar + blond peat (73:12:10:5; *v/v*), did not differ significantly from those grown in the coir + MSW + pine bark + blonde peat mix (Table 9). However, the shoot dry weight of the seedlings grown in coir + MSW + biochar + blond peat was lower than that grown in the mix coir + MSW + pine bark + blonde peat due to a higher allocation of biomass in the root system. This may indicate that seedlings in a mixture with biochar were subject to higher growth-constraining resources than those grown in the coir + MSW + pine bark + blonde peat. When nutrients are scarce, roots may allocate more biomass [61]. The two previous mixes had the highest seedling dry-matter accumulation, and their physicochemical characteristics were within an adequate range for substrate characteristics or slightly higher in the case of pH.

**Table 9.** Effect of the mix on seedling growth, experiment 2.

| | Fresh Weight | Dry Weight | | | | |
|---|---|---|---|---|---|---|
| Mixes[1] | Shoot | Shoot | Seedling | Seedling Dry Weight | Leaf Area | Leaves |
| | (g/Plant) | | | (%) | (cm$^2$) | (N°) |
| C + MSW + P | 1.96 b [2] | 0.13 b | 0.18 b | 4.65 a | 67.86 b | 6.25 a |
| C + MSW + BP | 2.12 b | 0.15 b | 0.20 b | 4.75 a | 68.83 b | 6.15 a |
| C + MSW + Pi | 2.38 ab | 0.15 b | 0.20 b | 4.44 a | 76.27 ab | 6.25 a |
| C + MSW + Pi + BP | 2.92 a | 0.20 a | 0.27 a | 4.80 a | 92.99 a | 6.69 a |
| C + MSW + B + BP | 2.49 ab | 0.16 b | 0.22 ab | 4.61 a | 81.04 ab | 6.73 a |
| Significance | * | * | * | NS | * | NS |

[1]—C—Coir, B—Biochar, MSW- Municipal solid organic waste, P—Perlite, Pi—Pine bark, BP—Blonde peat. [2]—Means followed by different letters within a column are significantly different * significant at $p < 0.05$ level. NS—not significant. Mean separation was performed using Duncan's multiple-range test.

The growth parameters do not correlate with the content of leaf photosynthetic pigments or total phenol. However, seedling total dry weight increased linearly with total shoot Chl (seedling total dry weight (g) = 0.466 × (total shoot Chl) + 0.0832, r$^2$ = 0.824,

$p < 0.01$) that was higher in mixes with four components. Seedling survival after transplanting is related to dry-weight accumulation. These results indicate that the mixes may use between 17 and 22% $v/v$ of locally produced renewable resources and are suitable for lettuce seedling growth.

The percentage of compost could be further increased when green raw materials are used. For instance, ref. [62] suggests the use of up to 50% compost as a component in a growing medium. Green waste compost is made from greenhouse vegetables, nursery shrubs, branches, plant trimmings, leaves, and grass from gardens, public green spaces, and other landscapes. The woody material is chopped, mixed with the remaining green residues, and gathered in clamps [4,63].

In the remaining mixes, the shoot fresh weight, shoot total dry weight, leaf area, and number of leaves did not differ significantly ($p < 0.05$) from those mixes grown in coir + MSW + biochar + blond peat (73:12:10:5; $v/v$). Despite the differences in seedling growth in different mixes, all seedlings from different treatments had well-developed shoot and root systems. The seedlings presented vigorous growth without any visual symptoms of deficiencies or toxicities. These results agree with those from [64]. Lettuce seedling growth in a fine-wood fiber substrate showed a good development in root mass and a lower leaf/root dry weight ratio, even by reducing the pot size. The pot size decreased, to some degree, the quality of lettuce seedling parameters. However, no differences in lettuce yield were found after transplanting to the field [64]. According to the authors, culture methods, such as, for instance, irrigation and good root development of seedlings in wood fiber substrates, have been responsible for these results. Thus, an adapted irrigation strategy to the substrate used plays a crucial role [65,66].

As previously mentioned, in addition to affecting biomass accumulation, in our study, the mixes also affected leaf TPC, which may influence seedling tolerance to abiotic stress after transplanting. Therefore, future research will be needed to assess the behavior of the seedlings after transplantation and compare their growth to that of seedlings grown on commercial substrates.

### 4. Conclusions

The findings of this study show that coir-based growing media with municipal solid waste compost and compost plus biochar can reduce the use of peat to a percentage of 5–10% $v/v$ and the use of 17–22% $v/v$ of locally produced renewable resources. The initial EC and pH of the mixes coir + MSW + pine bark + blonde peat (73:12:5:10 $v/v$) and coir + MSW + biochar + blonde peat (73:12:10:5, $v/v$) were within or were slightly higher than the maximal values of the range considered adequate for substrates. The presence of MSW (12%) and MSW (12%) + biochar (10%) in the mixtures had no adverse effects on seed germination and cumulative seed emergence. The seedling growth in those mixes was vigorous and higher than that of those grown in other mixtures. However, further research must compare lettuce seedling growth in these and commercial mixes and their use to grow other vegetable transplants. In addition, coir-based mixes affect total phenol content. As total phenol content increases tolerance to abiotic stress, future studies are needed to evaluate the behavior of the seedlings of the mixes after transplantation.

**Author Contributions:** R.M.A.M. conceived and designed the experiments; performed the experiments; analyzed and interpreted the data; contributed reagents, materials, analysis tools, or data; and wrote the paper. T.C.M. performed the experiments and analyzed the data. I.A.-P. and R.F. performed the experiments; analyzed and interpreted the data; contributed reagents, materials, analysis tools, or data; and wrote the paper. N.S.G. reviewed, corrected, and edited the paper. All authors have read and agreed to the published version of the manuscript.

**Funding:** This work is funded by National Funds through FCT—Foundation for Science and Technology under the Project UIDB/05183/2020.

**Data Availability Statement:** Not applicable.

**Conflicts of Interest:** The authors declare no conflict of interest.

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
