# Peer review of "Coir-Based Growing Media with Municipal Compost and Biochar and Their Impacts on Growth and Some Quality Parameters in Lettuce Seedlings"

_horticulturae, doi:10.3390/horticulturae9010105_

Round 1

Reviewer 1 Report

Dear authors,

the article is well ranged, which is important, and clearly define the aim. But after reading the text, I also consider that the aim (at the end of section 1) should include the interest to reduce the use of non-renewable source as it is the peat.

Nevertheless, some explanations could be given.

For example, It should be widely explained why 2 quite similar experiments are conducted to produce the results. Some of the trials (proportions between components) are similar in both.

The title refers to coir as a main point, but during the analysis of the results there are other things that seem to be more relevant, as the use of biochar of MSW compost.

For the rest, you can find especific comments in the pdf.

Author Response

Reviewer 1

Thank you for your valuable comments and suggestions. 

1 - The article is well ranged, which is important, and clearly define the aim. But after reading the text, I also consider that the aim (at the end of section 1) should include the interest to reduce the use of non-renewable source as it is the peat.

Amended: A sentence at the of the section 1 was introduced

Municipal compost and biochar are two locally produced renewable resources whose use lessens Portugal's dependency on peat and coir imports, keeps organic waste out of landfills, and reduces the carbon footprint [35] and greenhouse gas (GHG) emissions [12].

Amended: The sentence at the end of section 1 was changed to

“This research aimed to reduce the use of peat in substrates by investigating the suitability of coir-based substrates in combination with selectively harvested municipal solid organic waste and/or acacia biochar for successful horticultural plant cultivation.”

2- To address the question “For example, It should be widely explained why 2 quite similar experiments are conducted to produce the results. Some of the trials (proportions between components) are similar in both.

Except for mix (coir + MSW + perlite), the proportions of the components in the mixes of experiment 2 were very different from those used in experiment 1. A sentence was added to the manuscript to explain why. The sentence was:

“The EC in mixes with MSW was lower in the mixture (coir + MSW + perlite) (2.8 dS m-1) than in the other mixes. As a result of this, the change in the proportions of the components of this mixture in experiment 2 was reduced (Table 2).”

3 – For the rest, you can find specific comments in the pdf

Point-by-point response to reviewer 1's comments in PDF 

1-We agree: “growing-media” was amended to “coir based growing- media”

2-We disagree:  

In this MSW coming from separate collection or not? This this must be highlighted as it is important in content of pollutants

In the sentence "An alternative strategy could be to use selectively collected municipal solid organic waste," the use of selectively collected municipal compost is already mentioned, so we do not think it is necessary to amend the text.

In addition, in the material section, it was mentioned that the raw materials used in the manufacturing of compost "include horticultural products, food scraps carefully selected from restaurants, canteens, and similar establishments, forest exploitation residues (e.g., branches and foliage), and green residues (e.g., flowers, grasses, and pruning)."

3- We agree:  From the best of my knowledge MSW present low pH values due to rapid formation of acids. I would agree if you refer to compost of MSW

The sentence was amended to “In addition, MSW compost usually has high electrical conductivity and pH values [27].

4- We agree:

“The organic composts may also have human pathogens, which are potentially reduced when  raw materials are collected selectively, as in the present study”.

The sentence was amended to “The organic composts may also have human pathogens; to reduce them in the future, the compost must be certified as to the raw materials used and the maximum temperature and time of exposure to these during the thermophilic phase.”

In addition, in material and methods section was added the sentence “According to the manufacturer, the compost used in this study is free of pathogens.

5- We agree:

In the manuscript, "BT-blonde peat" was changed to "BP-blonde peat."

6- We agree:

“The mixture really loww from that of C+MSW+P showed in table 1. Which is the aim of having both”. To explain that, the following sentence was introduced in the manuscript

 “The EC in mixes with MSW was lower in the mixture (coir + MSW + perlite) (2.8 dS m-1) than in the other mixes. As a result of this, the change in the proportions of the components of this mixture in experiment 2 was reduced (Table 2).”

We agree:  These parameters are both phytochemical and physical. Please place it correct way

The sentence was amended  to “Moisture content, total porosity, and bulk density”

We agree:

We agree: “Aliquots of the methanolic extracts MW90 or MW80”

the sentence was amended clarifying the use of extract MW90:“The following equations were used to determine the concentration (mg/100 g FW) of chlorophyll a (Chl a), chlorophyll b (Chl b) and carotenoids (Cc) of MW90 extract, where A denotes absorbance, following [38]:

1.Chl a = 16.82A665.2 – 9.28A652.4

  1. Chl b = 36.92A652.4 – 16.54A665.2
  2. Cc = (1000A470 – 1.91Chl-a – 95.15Chl-b)/225

We agree: Does this really happen in the experiment ? Did you changed the mixtures of table 1 and 2 to achieve lower pH values?

The sentence was amended to “As a result, the ratios of the components in the mixes of experiment 2 (table 2) were altered to decrease the pH. Yes, we changed the proportions of components to reach better pH and EC values. The pH of the mixes in experiment 2 was slightly lower than experiment 1, probably due to the buffer power of the substrate.

We agree: Here it should be indicated how the extract should be prepared to compare with the results.  The sentence was amended to: to Martinez and Roca [44], the appropriate range of the EC for substrates ranges from 0.75 to 3.5 dS m-1, but they did not discriminate the method used to determine EC.”

We agree: table 4 In relation to your pertinent question about "mass wetness," Yes, we refer to dry substrate. To address your question, we added the table legend “Mass wetness—the water content of a sample on a dry mass basis. This is calculated as (wet weight − dry weight)/dry weight.”

We agree: “The seed emergence was high” the sentence was amended to  “The seed emergence was higher”

We agree: “Chl a, Chl b, and total Chl contents were high” the sentence was amended to “Chl a, Chl b, and total Chl contents were higher”

We agree:  “Fertilizer”  was changed to “fertilized”

We agree: “ low Ec value” was changed to “low EC value”

We agree: Is this a theoretical value, If so, please quote the reference. 

A reference was introduced “This may be due to the presence of humic acids, which account for 13% [34] of the dry weight of the compost, which was higher in these mixes (20%).

We agree:  “Fertilizer”  was changed to “fertilized”

We agree: table 7  In relation to your pertinent question about "mass wetness," Yes, we refer to dry substrate. To address your question, we added the table legend “Mass wetness—the water content of a sample on a dry mass basis. This is calculated as (wet weight − dry weight)/dry weight.”

Reviewer 2 Report

General comments

-This manuscript, by authors, studied

  “Coir-based media with municipal compost and biochar and their impacts on growth and some quality parameters of lettuce seedlings”.

Overall, the topic is of interest to horticulturae, readers. However, the following are the specific comments on the article concerning Major revision.

Specific Comments and Suggestions

-Abstract

-Too general

-using municipal compost and/or acacia biochar as components? Explain more

-No discussion about any environmental factors in details?

-Add more results in details.

-Significance of your study? Specific finding

-Introduction

-Introduction is not enough. Need more details and significance.

-Need to summarize and be specific with your concerned study.

Add more details for organic amendments methods in one or two sentence and between relationship for compost, vermicompost, biochar or biogas and greenhouse gasses emissions. May consider

“Technologies for municipal solid waste management: Current status, challenges, and future perspectives”

“Reuse of agricultural wastes, manure, and biochar as an organic amendment: A review on its implications for vermicomposting technology”

-Revise the introduction section with summarizing and significance of the study.

-Can add more references.

-Materials and Methods

-. Better to add tables of your two experiments with details.

-Methodology also needs to summarize and be specific.

-Unnecessary details make the manuscript confusing.

-Results and Discussion

-Too weak discussion. Please add other studies' references to support your current study and its significance

-Figures can be improved or increased in resolution. Not clear. Revise it, please.

-Conclusions

-Too simple

-“The findings of this work indicate that the mixes coir + MSW + pine bark + blonde peat (73:12:5:10 v/v) and coir + MSW + biochar + blonde peat (73:12:10:5, v/v) had an initial EC within the adequate range for substrates.”

-One or two paragraphs are appropriate for specific conclusions.

-Significant differences on which basis?

References

-Recheck reference format carefully. More references will be supportive to your study.

Author Response

Dear Reviewer,

Thank you for your comments. Please note that we had some difficulties and troubles understanding some of comments and suggestions. Below our answers.

General comments

-This manuscript, by authors, studied “

  “Coir-based media with municipal compost and biochar and their impacts on growth and some quality parameters of lettuce seedlings”.

Overall, the topic is of interest to horticulturae, readers. However, the following are the specific comments on the article concerning Major revision.

Specific Comments and Suggestions

-Abstract

-Too general

-using municipal compost and/or acacia biochar as components? Explain more

-No discussion about any environmental factors in details?

-Add more results in details.

-Significance of your study? Specific finding

Ameded. We improved the abstract according to your remarks.

-Introduction

-Introduction is not enough. Need more details and significance.

-Need to summarize and be specific with your concerned study.

Add more details for organic amendments methods in one or two sentence and between relationship for compost, vermicompost, biochar or biogas and greenhouse gasses emissions. May consider

“Technologies for municipal solid waste management: Current status, challenges, and future perspectives”

“Reuse of agricultural wastes, manure, and biochar as an organic amendment: A review on its implications for vermicomposting technology”

Thank you for your suggestion. We insert your suggestion for paper 2, but we do not see any reason to insert your first suggestion, because it miss the mark of our aim.

“-Revise the introduction section with summarizing and significance of the study.

Amended: The introduction was changed, and new sentences were added:

„Municipal compost and biochar are two locally produced renewable resources whose use lessens Portugal's dependency on peat and coir imports, keeps organic waste out of landfills, and reduces the carbon footprint [35] and greenhouse gas (GHG) emissions [12].

 We include a citation of the paper. " Raza, S.T., Wu, J., Rene, E.R., Ali, Z., & Chen, Z. (2022). Reuse of agricultural wastes, manure, and biochar as an organic amendment: A review on its implications for vermicomposting technology. Journal of Cleaner Production, 132200."

-Materials and Methods

- Better to add tables of your two experiments with details.

We fully agree with you suggestion. However, there are still tables with details, see table 1 and two.

-Methodology also needs to summarize and be specific.

-Unnecessary details make the manuscript confusing.

Taking into account the guide lines from the jorunal and the remarks of reviewer 1 and after reading the part „Methodology“ once again we realize that all details are necessary and we are pointed out all the important things.

-Results and Discussion

-Too weak discussion. Please add other studies' references to support your current study and its significance

Amended. We add more studies and references that support our results (plaese see the corrected manuscript).

-Figures can be improved or increased in resolution. Not clear. Revise it, please.

Amended: the figures were improved.

-Conclusions

Conclusions

Amended: The two first sentences of the conclusions were improved.

„The findings of this study show that coir-based growing media with municipal solid waste compost and compost plus biochar can reduce the use of peat to a percentage of 5 - 10% v/v and the use of 17 - 22% v/v of locally produced renewable resources. The initial EC and pH of the mixes coir + MSW + pine bark + blonde peat (73:12:5:10 v/v) and coir + MSW + biochar + blonde peat (73:12:10:5, v/v) were within or were slightly higher than the maximal values of the range considered adequate for substrates.“

-Significant differences on which basis?

The significance was based in statistical anailyses presnte results and discussion section

References

 -Recheck reference format carefully.

We double-checked the reference format that are now in agreement with the journal's author guidelines. 

More references will be supportive to your study.

Amended. Thank you; the following references are now included in the manuscript.

Raza, S.T.; Wu, J.; Rene, E.R.; Ali, Z.; Chen, Z. Reuse of agricultural wastes, manure, and biochar as an organic amendment: A review on its implications for vermicomposting technology. J Clean Prod 2022, 360. 132200. https://doi.org/10.1016/j.jclepro.2022.132200

Chrysargyris, A.; Prasad, M.; Kavanagh, A.; Tzortzakis, N. Biochar type. ratio. and nutrient levels in growing media affects seedling production and plant performance. Agronomy 2020, 10. 1421. https://doi.org/10.3390/agronomy10091421

Round 2

Reviewer 2 Report

The authors made reasonable efforts to improve the manuscript.

Figures can be improved before final publication. 

Author Response

Thank you for your comments.

Figure 1 has been improved.